Cell division cycle associated 2 (CDCA2) upregulation promotes the progression of hepatocellular carcinoma in a p53-dependant manner

Wang Jiahui 1
Liu Xin 1
Chu Hongjin 1
Chen Jian chenjianyt@163.com 2
1 Central Laboratory, The Affiliated Yantai Yuhuangding Hospital of Qingdao Unviersity , Yantai , Shandong , China
2 Department of Medical Oncology, The Affiliated Yantai Yuhuangding Hospital of Qingdao University , Yantai , Shandong , China
Gould Gwyn
Electronic publication date: 2022 Jun 6
Publication date: 2022
Volume: 10
Electronic Location ID: e13535
Received 2022 Feb 14; Accepted 2022 May 13
Copyright: ©2022 Wang et al.
Copyright year: 2022
Copyright holder: Wang et al.
License: This is an open access article distributed under the terms of the Creative Commons Attribution License, which permits unrestricted use, distribution, reproduction and adaptation in any medium and for any purpose provided that it is properly attributed. For attribution, the original author(s), title, publication source (PeerJ) and either DOI or URL of the article must be cited.
License URL: https://creativecommons.org/licenses/by/4.0/

Keywords: Hepatocellular carcinoma, Cell division cycle associated 2 protein, TCGA datasets, G1/S transition, Cell proliferation, Epithelial-mesenchymal transition, Cyclophilin J, P53, Glypican-3

Funding: Key Technology Research and Development Program of Shandong Province 2019GSF107096 Medical Health Science and Technology Development Program of Shandong Province 202102080625 This study was supported by the Key Technology Research and Development Program of Shandong Province (No. 2019GSF107096) and the Medical Health Science and Technology Development Program of Shandong Province (No. 202102080625). The funders had no role in study design, data collection and analysis, decision to publish, or preparation of the manuscript.

==============================
Background

Elevated expression and oncogenic functions of cell division cycle associated 2 (CDCA2), an important mitotic regulator, have been demonstrated in several cancer types, however their involvement in hepatocellular carcinoma (HCC) has not been elucidated, and the underlying molecular mechanism remains unclear. This study aims to determine the role of CDCA2 in HCC and the underlying molecular mechanism.

Methods

The expression of CDCA2 in HCC was studied in 40 pairs of frozen and 48 pairs of paraffin-embedded HCC samples and paracancerous normal samples by qRT-PCR and immunohistochemistry, respectively, and using The Cancer Genome Atlas (TCGA) datasets. The cellular function of CDCA2 was studied in vitro in the HepG2, Huh7 and SK-Hep1 HCC cell lines.

Results

We found significantly upregulated CDCA2 expression in HCC, which was correlated with higher clinical stage, tumor grade and Glypican-3 (+). High CDCA2 expression was correlated with worse overall survival. CDCA2 promoted the proliferation of HCC cells by promoting G1/S transition through the upregulation and activation of CCND1/CDK4/6 and CCNE1/CDK2, enhanced the clonogenic ability, inhibited apoptosis in a p53/p21-dependent manner by inhibiting the p38 MAPK pathway and activating the JNK/c-Jun pathway, and promoted the migration of p53-mutant Huh7 cells by activating the epithelial-mesenchymal transition. Targeting CDCA2 reduced the chemoresistance of HCC cells to cisplatin. CDCA2 expression was also regulated by cyclophilin J.

Conclusions

This study revealed elevated expression of CDCA2 in HCC, possibly as a result of p53 dysregulation, which was associated with worse prognosis of patients. We confirmed the oncogenic role of CDCA2 in HCC in vitro and revealed some of the underlying molecular mechanisms. This study indicated the potential value of CDCA2 as a future target for the treatment of HCC.

Introduction

Liver cancer is the 7th leading type of cancer worldwide, and China contributes approximately one-quarter of all new cases and related deaths each year (Bray et al., 2018). Hepatocellular carcinoma (HCC) comprises 75%–85% of cases of primary liver cancer (Bray et al., 2018). The tumorigenesis and development of HCC are multigene and multistep processes. Oncogenes, tumor suppressor genes, DNA repair genes, cell adhesion molecules, cell cycle regulators and the growth factor/receptor system are all involved in the development and progression of HCC. Cell division cycle associated 2 (CDCA2), originally known as Repo-Man, is a nuclear protein that recruits protein phosphatase 1γ to mitotic chromosomes and regulates the PP1-dependent DNA damage response during cell cycle progression (Trinkle-Mulcahy et al., 2006). The involvement of CDCA2 in cancer was first suggested in 2010, when elevated CDCA2 expression was preliminarily observed in 4 cases of liver cancer tissues (Peng et al., 2010). Then, the elevated expression and oncogenic functions of CDCA2 were demonstrated in several cancer types. High CDCA2 expression was associated with more advanced pathological features in clear-cell renal cell carcinoma (ccRCC), colorectal cancer (CRC), prostate cancer, oral squamous cell carcinoma (OSCC), lung adenocarcinoma, breast cancer, melanoma, and pancreatic ductal adenocarcinoma (Uchida et al., 2013; Wang et al., 2016; Shi et al., 2017; Feng et al., 2019; Zhang et al., 2020; Li et al., 2020; Chen et al., 2020; Jin et al., 2020). The cellular function of CDCA2 in cancerous cells has also been investigated in several cancer types, and CDCA2 is mainly involved in promoting tumor cell proliferation by preventing G1 phase arrest and inhibiting apoptosis (Uchida et al., 2013; Shi et al., 2017; Feng et al., 2019; Li et al., 2020). CDCA2 was also reported to be a metastatic biomarker of synovial sarcoma and to promote the migration of melanoma cells (Jin et al., 2020; De Necochea-Campion et al., 2017). The involvement of CDCA2 in HCC has not been elucidated, and the underlying molecular mechanism remains unclear.

In this study, we confirmed the elevated expression of CDCA2 in HCC and its correlation with poor prognosis. We report the pro-proliferation and anti-apoptosis effects of CDCA2 in HCC cells in vitro, discovered its regulation by cyclophilin J (CYPJ), and showed that its function involves activation of the epithelial-mesenchymal transition (EMT) process, the p53/p21 pathway, the p38 MAPK pathway and the JNK/c-Jun pathway.

Materials & Methods

Ethics approval

This study was approved by the Ethics Committee of Yantai Yuhuangding Hospital (IRB no.: YHD 2018-151). Written informed consent was obtained from all individual participants included in the study prior to study commencement for enrollment and for the use of patient clinical information for publication.

Molecular cloning

The CDCA2 overexpression plasmid was constructed by inserting the 3,072-bp coding sequence of CDCA2 (GenBank accession number: NM_152562.4) into the p3xFlag-CMV−7.1 vector. The CYPJ overexpression plasmid was constructed by inserting the 1178-bp coding sequence of PPIL3 (GenBank accession number: NM_032472.4) into the pCMV-myc vector. A GV493 vector containing three CDCA2-specific shRNA sequences was constructed to knock down CDCA2 (shCDCA2-1: gcAAACCTTTCAGAGGAGAAA, shCDCA2-2: gcCGTTCTCAGTTCTCCTAAT, shCDCA2-3: ccACAGTAACCGTAGAGCAAT).

RNA extraction and real-time quantitative PCR (qRT-PCR)

Pathologically confirmed specimens were collected during surgery from patients treated at Yantai Yuhuangding Hospital (Shandong, China), total RNA was extracted, and qRT-PCR analysis was performed on an FTC-3000P Real-Time Quantitative Thermal Cycler (Funlyn, Canada) as previously described (Wang et al., 2021). All the experiments were performed at least in triplicate. The primer sequences used for qRT-PCR were: CDCA2-F: 5′-AAAGGAGGAACACCTGTTTGTA-3′, CDCA2-R: 5′-CCTGAAAGGGTCTCAGAGATTG-3′, β-Tubulin-F: 5′-GGACCGCATCTCTGTGTACTA-3′, β-Tubulin-R: 5′-CTTTGGCCCAGTTGTTACCT- 3′, p21-F: 5′-GTGGACCTGTCACTG TCTTG- 3′, p21-R: 5′-TTAGGGCTTCCTCTTGGAGA- 3′, TP53-F: 5′-GGAAGAGAAT CTCCGCAAGAAA- 3′, TP53-R: 5′-CTCATTCAGCTCTCGGAACATC- 3′, CYPJ-F: 5′-GGAAGAGGAGGCAACAGTATTT- 3′, CYPJ-R: 5′-GGCAACTTCTCCAACTCATCTA-3′.

Western blotting

Total proteins were extracted from cells lysed with RIPA lysis buffer (Beyotime, China) and subjected to Western blotting analysis using rabbit/mouse primary antibodies from Affinity Biosciences, USA [anti-CDCA2 (DF3543), anti-Cyclin E1 (AF6235), anti-GAPDH (T0004), anti-β-Actin (T0022), anti-β-Tubulin (T0023)], or Cell Signaling Technology, USA [Cell Cycle Regulation Antibody Sampler Kit (9932T), Stress and Apoptosis Antibody Sampler Kit (8357T), Epithelial-Mesenchymal Transition (EMT) Antibody Sampler Kit (9782T)], and HRP-conjugated goat anti-rabbit/mouse secondary antibodies (ZSGB-BIO, China). The immunoblots were developed with ECL reagent (Thermo Scientific, USA). Densitometric analysis was performed to determine the grayscale intensity of the target bands using ImageJ software (imagej.nih.gov/ij).

Immunohistochemistry (IHC)

IHC was performed as previously described (Lin et al., 2017) except that HRP-conjugated secondary antibodies (ZSGB-BIO, China) were used for staining and a DAB Kit (ZSGB-BIO, Beijing, China) was used for color development.

Cell culture and transfection

HCC cell lines were obtained from iCell Bioscience (China). The cells were routinely cultured in MEM (iCell Bioscience, Shanghai, China) for HepG2 and SK-Hep1 cells or in DMEM (Biological Industries, Beit-Haemek, Israel) containing nonessential amino acids (Invitrogen, Waltham, MA, USA) for Huh7 cells, and the media were supplemented with 10% fetal bovine serum (Biological Industries, Beit-Haemek, Israel). The cells were cultured at 37 °C with 5% CO2 humidified air according to standard procedures.

Transient plasmid transfection was performed with XtremeGENE HP DNA Transfection Reagent (Roche, Switzerland) following the manufacturer’s instructions.

Lentivirus-mediated CDCA2 knockdown

Huh7 and SK-Hep1 cells were both infected with lentivirus carrying the GV493 vector with CDCA2-specific shRNA (shCDCA2, target sequence: gcAAACCTTTCAGAGGAGAAA) or a nontargeting control sequence (shNC, sequence: TTCTCCGAACGTGTCACGT) at MOIs of 5 and 30, respectively, and uninfected cells were used as the control. After three days of growth, the cells were observed with a fluorescence microscope to confirm the expression of EGFP in >90% of cells. Puromycin dihydrochloride (MedChemExpress, Monmouth Junction, NJ, USA) was added to the complete growth media at a final concentration of 8 µg/mL for Huh7 cells and 6 µg/mL for SK-Hep1 cells, and this medium was refreshed every 3 days until all the uninfected cells had died and no more death occurred in the infected cells. The cells were then digested and seeded in 96-well plates at a concentration of 1 cell/well and continued to grow until stable strains were obtained.

CCK-8 assay

The proliferation of cells was measured using Cell Counting Kit-8 (Dojindo Laboratories, Japan) as previously described (Wang et al., 2021).

Cell viability assays were performed by treating the cells (5 × 103/well) with cisplatin (CDDP) in DMSO for 48 hrs. An equal volume of DMSO was added to the control wells. Each sample was analyzed in triplicate. The OD450 was measured after 2 hr of incubation at 37 °C with 100 µL serum-free MEM containing 10 µL CCK-8 solution. Cell viability = ODsample/ODcontrol × 100%.

Transwell migration assay

Transwell migration assays were performed as previously described (Yang et al., 2020) with 4 × 104 cells seeded per 8-µm Transwell chamber (Corning, Corning, NY, USA).

Colony formation assay

Colony formation assays were performed as previously described (Yang et al., 2020).

Tunel

TUNEL assay was performed as previously described (Wang et al., 2021).

Flow cytometry

Flow cytometry analysis of cell cycle and apoptosis was performed on a Moflo XDP flow cytometer (Beckman Coulter, Brea, CA, USA) using the Cycletest™ Plus DNA Reagent Kit (BD Biosciences, USA) and FITC Annexin V Apoptosis Detection Kit I (BD Biosciences, San Jose, CA, USA).

Bioinformatics analysis

Clinical data and RNA-seq v2 data from The Cancer Genome Atlas (TCGA) database (http://cancergenome.nih.gov/) were retrieved from cBioPortal (http://www.cbioportal.org/) and analyzed. The CDCA2 transcription level and promoter methylation level in HCC tissues were compared using the UALCAN database (http://ualcan.path.uab.edu/) (Chandrashekar et al., 2017). Gene function enrichment analysis was performed using the Gene Ontology Consortium (GO) (http://geneontology.org/).

Statistical analysis

Statistical analysis was performed using GraphPad Prism 8.0 software. The log-rank (Mantel-Cox) test was used to compare Kaplan–Meier survival curves. Numerical data with a normal distribution were compared by Student’s t-test for two samples or one-way ANOVA for three or more samples. Categorical data were compared with Fisher’s exact test. Ratios were compared by Kruskal–Wallis test for two samples or Mann–Whitney test for three or more samples.

Results

Elevated CDCA2 expression in HCC was associated with malignant features and worse prognosis

CDCA2 expression in HCC tissues and patient survival were first analyzed using RNA-seq data from UALCAN. The results showed a significantly higher transcription level of CDCA2 in HCC tissues (n = 371) than in normal tissues (n = 50) (P < 0.0001, Fig. 1A), and CDCA2 expression increased with clinical stage (Fig. 1B). Interestingly, CDCA2 expression was significantly higher in patients expressing mutant TP53 than in patients expressing wild-type TP53 (P < 0.0001) (Fig. 1C). The survival rate of the high CDCA2 group was worse than that of the low CDCA2 group in the complete dataset, which included all ethnic groups (P < 0.0001), and in Asian patients (P < 0.0001), but this trend was not observed in White Caucasian patients (P = 0.3846) (Figs. 1D and 1E).

Figure 1 Expression of CDCA2 in hepatocellular carcinoma.

Analysis of an RNA-seq dataset from UALCAN showed (A) elevated CDCA2 expression in primary HCC tissues compared with normal liver tissues; (B) increased CDCA2 levels with clinical stage; (C) elevated CDCA2 expression in samples expressing mutant TP53 compared with samples expressing wild-type TP53; (D) worse survival of patients with high CDCA2 expression than in patients with low CDCA2 expression; and (E) worse survival of patients with high CDCA2 expression than patients with low CDCA2 expression among Asian patients but not among White Caucasian patients. (F) Our study found elevated CDCA2 mRNA levels in 40 HCC tissues compared to paired normal liver tissues. (G) Representative photos of +ve and –ve immunohistochemistry staining of CDCA2 in HCC or normal liver tissues. (H) qRT-PCR measured CDCA2 mRNA expression in the L02 and HCC cell lines. (I) Western blotting analysis of CDCA2 protein expression in the L02 and HCC cell lines. Stars indicate the P value of the CDCA2 protein level compared with that in the L02 cells. (J) Western blotting analysis of CDCA2 protein expression in HCC cell lines with lentivirus-mediated stable CDCA2 knockdown. *, P < 0.05; **, P < 0.01; ***, P < 0.001; ****, P < 0.0001.

The correlations between CDCA2 mRNA expression and the clinicopathological characteristics of HCC patients were analyzed using the TCGA LIHC_pan_caner_atlas_2018 dataset. The results suggested correlations between high CDCA2 expression and lower age (P = 0.0213), advanced clinical stage (P = 0.0001), and higher tumor grade (P = 0.0011). In addition, CDCA2 expression in Asian patients was significantly higher than that in non-Asian patients (P = 0.0405), and CDCA2 was more frequently mutated in the high expression group (P = 0.0081) (Table 1).

Table 1 Correlations between CDCA2 mRNA expression and clinicopathological characteristics of hepatocellular carcinoma patients from the TCGA LIHC_pan_caner_atlas_2018 dataset.

Characteristics	N	High CDCA2 expression, N (%) (N = 183)	Low CDCA2 expression, N (%) (N = 183)	P	
Age				0.0213*	
≤60	176	99 (56.3)	77 (43.7)		
>60	189	83 (67.4)	106 (32.6)		
Sex				0.658	
Male	120	64 (53.3)	56 (46.7)		
Female	256	129 (50.4)	127 (49.6)		
Clinical stage				0.0001***	
I–II	256	114 (44.5)	142 (55.5)		
III–IV	91	62 (68.1)	29 (31.9)		
Tumor grade				0.0011**	
G1–G2	225	97 (43.1)	128 (56.9)		
G3–G4	136	83 (61.0)	53 (39.0)		
Race				0.0405*	
Asian	166	101(60.8)	65 (39.2)		
Non-Asian	196	86 (43.9)	110 (56.1)		
Mutation count		116.6 ± 145.5	86.37 ± 45.78	0.0081**	

We confirmed the mRNA expression level of CDCA2 in 40 pairs of HCC tissues and normal liver tissues by qRT-PCR, and the results showed higher expression in HCC tissues than in normal tissues (P < 0.001) (Fig. 1F). CDCA2 expression was also studied in 48 paraffin-embedded HCC tissues by IHC (Fig. 1G). The correlations between CDCA2 protein expression and the clinicopathological characteristics of the patients were analyzed, and the results showed an association of high CDCA2 protein expression with high tumor grade (P = 0.0405) and Glypican-3 positivity (P = 0.049) (Table 2). Although not statistically significant, there were higher percentages of high-CDCA2 samples in groups with advanced clinical stage and larger tumor size.

Table 2 Correlations between CDCA2 protein expression and clinicopathological characteristics in paraffin-embedded hepatocellular carcinoma samples.

Characteristics	N	High CDCA2 expression, N (%) (N = 21)	Low CDCA2 expression, N (%) (N = 27)	P	
Age (yrs)				0.383	
≤60	25	9 (36.0)	16 (64.0)		
>60	23	12 (52.2)	11 (47.8)		
Sex				0.153	
Male	41	15 (36.6)	24 (63.4)		
Female	9	6 (66.7)	3 (33.3)		
Clinical stage				0.380	
I–II	30	11 (36.7)	18 (63.3)		
III–IV	19	10 (52.6)	9 (47.4)		
Tumor grade				0.0405*	
G1–G2	25	7 (28.0)	18 (72.0)		
G3	23	14 (60.9)	9 (39.1)		
Tumor size (cm)				0.0798	
≤4	24	7 (29.2)	17 (70.8)		
>4	24	14 (58.3)	10 (41.7)		
Glypican-3				0.049*	
−	13	4 (30.8)	9 (69.2)		
+	35	23 (68.6)	12 (31.4)		

CDCA2 knockdown and overexpression in HCC cell lines

The levels of CDCA2 in the L02 liver cell line and three HCC cell lines, namely, Huh7, SK-Hep1 and HepG2 cells, were detected by qRT-PCR and Western blotting (Figs. 1H and 1I). All three HCC cell lines had significantly higher CDCA2 expression than the L02 cell line. CDCA2 expression was highest in Huh7 cells and lowest in HepG2 cells. Therefore, CDCA2 was transiently overexpressed in HepG2 cells (HepG2-pCDCA2), and cells transfected with an empty vector served as a control (HepG2-p3xFlag). Three CDCA2-specific shRNAs were transfected into Huh7 cells to achieve transient knockdown of CDCA2. shCDCA2-1 and shCDCA2-2 appeared to have similar knockdown efficiency, both better than shCDCA2-3 (Fig. 1I), and shCDCA2-1was packaged into lentivirus to achieve the stable knockdown of CDCA2 in the Huh7 and SK-Hep1 cell lines (Lv-shCDCA2) for subsequent functional and mechanistic studies, and cells transduced with nontargeting sequences served as a control (Lv-shNC). The alteration of CDCA2 expression was confirmed by qRT-PCR and Western blotting (Figs. 1H and 1J).

CDCA2 promoted HCC cell proliferation and migration in vitro

A CCK-8 assay was used to investigate the role of CDCA2 in the proliferation of HCC cells. The results showed that knockdown of CDCA2 significantly attenuated the growth of Huh7 and SK-Hep1 cells, while overexpression of CDCA2 promoted the proliferation of HepG2 cells (Fig. 2A). Cell clonogenic ability was also studied, and the results suggested that CDCA2 overexpression led to an increased number of colonies formed by HepG2 cells (P = 0.0136) (Fig. 2B). These results confirmed the pro-proliferative role of CDCA2 in HCC cells.

Figure 2 CDCA2 promoted HCC cell proliferation, migration, and G1/S phase transition of HCC cells in vitro.

(A) Proliferation of CDCA2-knockdown Huh7 or SK-Hep1 cells and CDCA2-overexpressing HepG2 cells was measured by CCK-8 assay. (B) Colony formation assay showed increased clonogenic ability of HepG2 cells after overexpression of CDCA2. (C) Transwell migration assays showed that knockdown of CDCA2 attenuated the migratory ability of Huh7 cells. (D) Western blotting analysis of epithelial-mesenchymal transition marker proteins in HCC cells with CDCA2 knockdown or overexpression. (E) Flow cytometry analysis showed the promotion of the G1-S phase transition in CDCA2-upregulated HepG2 cells and the inhibition of the G1-S transition in CDCA2-downregulated Huh7 cells. (F) The protein levels of CCNDs and Cdks in HCC cells with CDCA2 knockdown or overexpression were measured by Western blotting. (G) mRNA expression of p21 and p53 in HCC cells with CDCA2 knockdown or overexpression was measured by qRT-PCR. *, P < 0.05; **, P < 0.01; ***, P < 0.001; ****, P < 0.0001.

We then investigated the role of CDCA2 in HCC cell migration by transwell assay and found reduced migration capacity after CDCA2 was knocked down in Huh7 cells (Fig. 2C). However, cell migration was not significantly affected in SK-Hep1-Lv-shCDCA2 or HepG2-pCDCA2 cells compared to their controls.

We then investigated the expression of key EMT markers in HCC cells using Western blotting. The results showed increased levels of E-cadherin and decreased levels of Vimentin and the E-cadherin suppressor ZEB-1 in CDCA2-knockdown Huh7 cells compared to control cells, suggesting inhibition of the EMT pathway, and increased levels of N-cadherin in CDCA2-overexpressing HepG2 cells, indicating increased mesenchymal features. The levels of EMT markers were unchanged, or not expressed, in CDCA2-knockdown SK-Hep1 cells (Fig. 2D).

CDCA2 promoted the G1/S transition of HCC cells possibly by the upregulation and activation of CCND1/CDK4/6 and CCNE1/CDK2

The cell cycle progression of the three HCC cell lines was studied by flow cytometry. After CDCA2 knockdown, an increase in the proportion of Huh7 cells in the G1 phase was observed, whereas CDCA2 overexpression caused a decrease in the proportion of HepG2 cells in the G1 phase and an increase in the S phase (Fig. 2E). CDCA2 knockdown did not significantly affect the cell cycle progression of SK-Hep1 cells. These data suggested that CDCA2 could promote the transition from the G1 phase to the S phase in the cell cycle.

We examined the expression of key components of cell cycle regulation, cyclins (CCNs) and cyclin-dependent kinases (CDKs), by Western blotting. The results showed reduced levels of CCND1/CCNE1/CDK2/CDK4 in CDCA2-knockdown Huh7 cells, reduced levels of CCND1/CCNE1 in CDCA2-knockdown SK-Hep1 cells and increased levels of CCND1/CCND3/CCNE1/CDK2/CDK6 in CDCA2-overexpressing HepG2 cells (Fig. 2F). We also explored the mRNA expression of cyclins and CDKs in the TCGA LIHC_pan_caner_atlas_2018 dataset, which includes 366 patients, and found that CDCA2 expression was strongly or moderately correlated with the expression of CCNE1 (r = 0.635, P < 0.0001), CDK2 (r = 0.552, P < 0.0001) and CDK4 (r = 0.547, P < 0.0001) but was not correlated with the expression of CCND1, CCND3 and CDK6. Taken together, our results suggested that CDCA2 might promote the G1/S phase transition of HCC cells in part by upregulating CCND1/CDK4/6 and CCNE1/CDK2 expression.

To further understand the mechanism by which CDCA2 promotes the growth of HCC, the expression of the CDK inhibitors p21, p27, and p18 in HCC cells was detected by qRT-PCR and/or Western blotting. We observed upregulation of p18 and p27 expression and a lack of p21 expression in Huh7-Lv-shCDCA2 cells and the upregulation of p21 expression and downregulation of p18 and p27 expression in HepG2 cells (Figs. 2F and 2G).

CDCA2 played an antiapoptotic role in HCC cells by inactivating the p38 MAPK pathway and activating the JNK/c-Jun pathway

The apoptosis of CDCA2-overexpressing HepG2 cells was studied by flow cytometry. Apoptosis was induced by treatment with 5 mg/L CDDP for 24 hrs. In both the CDDP-treated and untreated groups, CDCA2 overexpression protected HepG2 cells from apoptosis, especially early apoptosis (Fig. 3A). The apoptosis of CDCA2-knockdown Huh7 and SK-Hep1 cells was investigated by TUNEL staining. Apoptosis was induced by treatment with 10 mg/L and 20 mg/L CDDP for 24 hrs. The results showed that Huh7 and SK-Hep1 cells were not as sensitive to CDDP-induced cell death as HepG2 cells. In particular, the apoptosis of Huh7 cells was not induced by CDDP at concentrations up to 20 mg/L, but this resistance was abolished by CDCA2 knockdown (P = 0.0418) (Fig. 3B). Treatment with 20 mg/L CDDP induced the apoptosis of SK-Hep1 cells (P = 0.0006 for the control group, P = 0.0049 for the CDCA2-knockdown group), and CDCA2 knockdown enhanced the chemosensitivity of SK-Hep1 cells to CDDP (P = 0.0435) (Fig. 3C).

Figure 3 Anti-apoptosis effect of CDCA2 in HCC cells.

(A) Flow cytometry analysis showed reduced apoptosis, especially reduced early apoptosis, of HepG2 cells after CDCA2 overexpression with or without treatment with 5 mg/L cisplatin (CDDP). TUNEL staining showed that (B) CDCA2 knockdown sensitized Huh7 cells to apoptosis induced by 20 mg/L CDDP and (C) led to increased apoptosis of SK-Hep1 cells after treatment with 20 mg/L CDDP. (D) The protein levels of stress and apoptosis markers in HCC cells with CDCA2 knockdown or overexpression were measured by Western blotting. *, P < 0.05; **, P < 0.01; ***, P < 0.001.

Western blotting analysis of stress and apoptosis markers showed no change in the Caspase 3/cleaved PARP and p38 MAPK/MAPKAPK2 pathways and inactivation of the JNK/c-Jun and p53 pathways in Huh7-Lv-shCDCA2 cells compared to the control cells. In SK-Hep1-Lv-shCDCA2 cells, there were increased levels of Caspase 3/cleaved PARP, activation of the p38 MAPK/MAPKAPK2 pathway, and inhibition of the JNK/c-Jun and p53 pathways. In CDCA2-overexpressing HepG2 cells, we found activation of p53 and JNK/c-Jun and inactivation of p38 MAPK (Fig. 3D). The data described above suggested that CDCA2 plays antiapoptotic and proliferative roles, possibly through the inactivation of the p38 MAPK pathway and the activation of the JNK/c-Jun pathway in HCC cells.

GO analysis of genes correlated with CDCA2 in HCC

Genes correlated with the expression of CDCA2 were pooled from cBioPortal, and of these genes, 386 genes with a Pearson’s correlation coefficient of ≥0.6 were analyzed by GO for functional enrichment analysis. The enriched molecular functions among these proteins included DNA binding, methylated histone binding, protein kinase activator activity, microtubule binding, cyclin-dependent protein Ser/Thr kinase activity, etc. The pathways positively enriched among these genes were DNA replication (19.37-fold enrichment, FDR = 2.36E−08) and p53 pathways (6.27-fold enrichment, FDR = 8.11E−04). The enriched reactome pathways included mitotic metaphase/anaphase transition, G2/M DNA replication checkpoint, mitotic G1 phase and G1/S transition, cyclin A/B1/B2-associated events during G2/M transition, CHK1/CHK2 (Cds1)-mediated inactivation of the cyclin B:CDK1 complex, TP53 regulation of cell cycle gene transcription, regulation of TP53 activity through phosphorylation, etc.

CDCA2 expression was regulated by CYPJ

Our research group previously discovered that the peptidyl-prolyl cis/trans-isomerase (PPIase) CYPJ could promote the growth of HCC by preventing G1 phase arrest by activating the cyclin D1 promotor (Chen et al., 2015). We therefore investigated the possible relationship between CDCA2 and CYPJ in HCC cells. We measured the mRNA and protein levels of CDCA2 and CYPJ in three HCC cell lines with altered CDCA2 or CYPJ expression (Figs. 4A and 4B). Our results showed no change in the CYPJ protein levels in CDCA2-knockdown or CDCA2-overexpressing HCC cells. However, overexpressing CYPJ significantly upregulated the expression of CDCA2. These results suggested that CYPJ might be a regulator of CDCA2.

Figure 4 CDCA2 was regulated by CYPJ.

(A) mRNA levels of CDCA2 and CYPJ in HCC cells with altered CDCA2 expression were detected by qRT-PCR. (B) Protein levels of CDCA2 and CYPJ in HCC cells with altered CDCA2 expression were detected by Western blotting. (C) The CYPJ inhibitor CsA caused the death of HCC cells in a dose-dependent manner, and CDCA2 overexpression alleviated this cytotoxic effect. (D) CDCA2 overexpression resulted in elevated cell viability of CsA-treated HCC cells as measured by the CCK-8 assay.

We then compared SK-Hep1 and HepG2 cells overexpressing CDCA2 with control cells treated with different concentrations of cyclosporine A (CsA), a PPIase inhibitor of most cyclophilins. The results showed that CDCA2 could alleviate the inhibition of cell growth caused by reduced cyclophilin activity (Figs. 4C and 4D).

Discussion

In this study, we validated, for the first time, the upregulated expression of CDCA2 in HCC and identified the correlation between higher CDCA2 expression and more advanced clinical stage and tumor grade, Glypican-3 (+), and worse overall survival; these results suggested the worse prognosis of patients with high CDCA2 expression. Interestingly, CDCA2 expression was higher in Asian patients, and the difference in overall survival between patients with high CDCA2 expression and low CDCA2 expression was most significant in Asian patients, while there was no difference in White Caucasian patients. This could possibly be due to a high CDCA2 expression profile in the Asian population originating from genetic or environmental factors and might contribute to the higher incidence of HCC in Asian countries, especially China, than in other parts of the world (Bray et al., 2018).

We confirmed that CDCA2 could promote the proliferation and colony formation of HCC cells, as in several other types of tumor cells, by facilitating G1/S transition while inhibiting cell apoptosis. We also observed that knockdown of CDCA2 expression led to reduced migratory ability of the Huh7 cell line due to downregulation of the EMT pathway. It is worth noting that SK-Hep1 cells have an endothelial origin and morphological characteristics (Tai et al., 2018) and therefore lack the epithelial marker E-cadherin, as shown in Fig. 2D. This explains the results of our transwell assay that the migration of SK-Hep1 cells was not affected by CDCA2.

CDCA2 regulates cell cycle progression in different cancers in part by affecting the expression of cell cycle regulatory molecules. In OSCC, knockdown of CDCA2 led to reduced levels of CDK4/6, CCND1 and CCNE1 (Uchida et al., 2013). In ccRCC and CRC, CCND1 was downregulated in CDCA2-knockdown cells, whereas CCNE1 was not (Feng et al., 2019; Li et al., 2020). In lung adenocarcinoma cells, the expression of CCNE1, but not CCND1, was positively correlated with the CDCA2 levels (Shi et al., 2017). Our study found that the CDCA2 expression level was correlated with that of CCND1/CCNE1/CDK2/4/6 in HCC cells, although only CCNE1, CDK2 and CDK4 were positively correlated with CDCA2 expression in the TCGA dataset. As not all cell cycle regulatory proteins have been examined in previous studies, we believe that the expression of different CCNs and CDKs is regulated by CDCA2 depending on the cancer type.

Cell cycle progression is controlled by the Cip/Kip family of CDK inhibitors and the INK4 family (inhibitors of CDK4). Although p21 (Waf1/Cip1) and p27 (Kip1) are potent inhibitors of cyclin E-dependent CDK2, they can also act as positive regulators of cyclin D-dependent CDK4/6 (Cheng et al., 1999; Sherr & Roberts, 1999). In our study, the expression of p18 (INK4) and p27 (Kip1) was upregulated in CDCA2 knockdown cells and downregulated in CDCA2-overexpressing cells. Therefore, the arrest of G1/S transition in CDCA2-knockdown Huh7 cells might be attributed to both downregulation of CCND1/CCNE1/CDK2/CDK4 expression and enhanced inhibition of CDK2/4 activities by upregulated p18 and p27 expression. On the other hand, CDCA2 overexpression in HepG2 cells led to increased p21 expression and decreased p18 and p27 expression, which might facilitate the assembly of CCN/CDK complexes in addition to upregulation of CCND/CCNE1/CDK2/6 expression, thus promoting the G1/S transition.

In this study, we found that the elevated expression and oncogenic function of CDCA2 was closely related to loss of normal p53 function and the p53/p21 pathway. First, bioinformatics analysis revealed higher CDCA2 expression in samples expressing mutant TP53 than in samples expressing wild-type TP53. Second, CDCA2 expression was correlated with enrichment of p53 pathways and the reactome pathways “TP53 Regulates Transcription of Cell Cycle Genes” and “Regulation of TP53 Activity through Phosphorylation”. Third, among the three HCC cell lines, CDCA2 was most highly expressed in Huh7 cells, which expressed mutant p53 and lacked p21 expression (Arima et al., 2002). Furthermore, among the three HCC cell lines, Huh7 cells had the lowest apoptosis rate and were resistant to CDDP-induced cell death, as p53-dependent apoptosis modulates the cytotoxicity of anticancer agents, such as ionizing radiation, 5-fluorouracil, etoposide, and adriamycin (Lowe et al., 1993), and p53 defects promote CDDP resistance in HCC cells (Li et al., 2018). In our results, inhibition of CDCA2 sensitized Huh7 cells and enhanced the sensitivity of SK-Hep1 and HepG2 cells to CDDP-induced apoptosis, suggesting a potential therapeutic value of targeting CDCA2 in reducing the chemoresistance of HCC to cisplatin. Moreover, CDCA2 showed antiapoptotic activity in SK-Hep1 and HepG2 cells but not in Huh7 cells, suggesting that its protection against apoptosis might be dependent on the p53/p21 pathway. In addition, Huh7 cells were also the only cell line in which CDCA2 activated EMT and promoted cell migration. This might also be attributed to p53 mutation in Huh7 cells. Mut-p53 has been reported to induce EMT and participate in the invasion and metastasis of endocrine carcinomas (Li et al., 2019). It is possible that mut-p53 plays the same role in HCC. We speculated that CDCA2 might have a synergetic effect with mut-p53 in inducing EMT and cell migration, although this assumption still requires further investigation.

This study also identified the possible regulation of CDCA2 by CYPJ, as its overexpression led to increased CDCA2 levels, whereas CYPJ expression was not affected by the CDCA2 levels. CYPJ can upregulate the transcription of a series of crucial genes relevant to cell cycle progression, including E-box, E2F, retinoblastoma, p53, activator protein 1, and CCND1 (Chen et al., 2015; Chen et al., 2016). Therefore, CDCA2 might be an important participant and coregulator in the oncogenic regulation of HCC by CYPJ.

Conclusions

In conclusion, this study found mut-p53-induced upregulation of CDCA2 expression in HCC, which was correlated with malignant features and worse patient’s prognosis. CDCA2 promoted the proliferation of HCC cells by promoting G1/S transition through the upregulation and activation of CCND1/CDK4/6 and CCNE1/CDK2, inhibited apoptosis by inactivating the p38 MAPK pathway and activating the JNK/c-Jun pathway in a p53/p21-dependent manner, and promoted the migration of Huh7 cells by activating the EMT pathway in vitro. We also identified the potential therapeutic value of targeting CDCA2 in reducing the chemoresistance of HCC to cisplatin.

Supplemental Information

Supplemental Information 1 Original images of Western blot

Click here for additional data file.

Data S1 Raw data for qRT-PCR and cellular assays

Click here for additional data file.

Table S1 Raw data for Table 1

Click here for additional data file.

Table S2 Raw data for Table 2

Click here for additional data file.

Additional Information and Declarations

Competing Interests

Author Contributions

Human Ethics

Data Availability

The authors declare there are no competing interests.

Jiahui Wang performed the experiments, analyzed the data, prepared figures and/or tables, authored or reviewed drafts of the article, and approved the final draft.

Xin Liu performed the experiments, analyzed the data, authored or reviewed drafts of the article, and approved the final draft.

Hongjin Chu performed the experiments, authored or reviewed drafts of the article, and approved the final draft.

Jian Chen conceived and designed the experiments, authored or reviewed drafts of the article, and approved the final draft.

The following information was supplied relating to ethical approvals (i.e., approving body and any reference numbers):

Ethics Committee of Yantai Yuhuangding Hospital granted Ethical approval to carry out the study within its facilities (IRB no.: YHD 2018-151).

The following information was supplied regarding data availability:

Data is available at Zenodo, DOI: 10.5281/zenodo.5904997.

Link: https://zenodo.org/record/5904997#.YfDs4km-uM8.

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
