# Peer review of "Cell division cycle associated 2 (CDCA2) upregulation promotes the progression of hepatocellular carcinoma in a p53-dependant manner"

_PeerJ, doi:10.7717/peerj.13535_

## Round 0.1 · original submission · Minor Revisions

As you will see from the comments below, the reviewers and I found your work robust, interesting and rigorous. Some minor comments on methodology and content are suggested below. Please address each of these carefully and provide a detailed response to all points raised in a rebuttal letter.

I would urge you to carefully proofread the manuscript for both typos and grammar. There are some places where an improvement in English would clarify and enhance your work.

Thank you for the submission and I look forward to hearing from you with the rebuttal and revised paper.

·

Basic reporting

This study aims to determine the role of cell division cycle associated 2 (CDCA2) in hepatocellular carcinoma (HCC) and its underlying molecular mechanism. The background is sufficient. The English need some revision. The figures, tables and images are of high quality. It is self-contained with relevant results regarding the hypothesis.

Experimental design

The scientific method is rigorous, the objectives are clear, it is well explained and justified, the experimental procedures are modern and sound, the experimental design is appropriate, replicates were included, they selected appropriate controls, and the results are interesting. The figures and images are of high quality, and they do not seem to be manipulated.

Validity of the findings

This study revealed elevated expression of CDCA2 in HCC, possibly because of p53 dysregulation, which was associated with worse prognosis of patients. They confirmed the oncogenic role of CDCA2 in HCC in vitro and revealed some of the underlying molecular mechanisms. This study indicated the potential value of CDCA2 as a future target for the treatment of HCC. Conclusion are clear and discussed properly.

Additional comments

1. The title must be corrected to: “Cell division cycle associated 2 (CDCA2) upregulation promotes the progression of hepatocellular carcinoma in a p53-dependent manner”
2. In Abstract. Instead of: “...had been demonstrated in several cancer types, however its involvement…” I suggest: “...have been demonstrated in several cancer types, however their involvement...”
3. In Abstract. Instead of: “…and using TCGA datasets.” I suggest: “…and using the Cancer Genome Atlas (TCGA) database.
4. In Introduction, line 72: “We report…”
5. In RNA extraction and real time PCR (qRT-PCR), line 103: “The mRNA expression levels of the target genes were compared by 2−ΔΔCt” I did not find this result in the Figures or Tables.
6. In line 235: I did not grasp the difference between Figure 1I with Figure 2G. Both seem related to measures of knockdown efficiency.
7. In Discussion, ¿What does OSCC stand for?
8. In Discussion, line 383: omit “were”
9. In Discussion, line 396: “CDCA2 might have a synergetic effect with mut-p53 in 397 inducing EMT and cell migration.” I do not see from what result(s) it is inferred a synergistic effect. Beside mut-p53 was reported for endocrine carcinomas. ¿Can it be extrapolated to hepatocellular carcinoma?

Reviewer 2 ·

Basic reporting

This paper presented a study about mut-p53-induced upregulation of CDCA2 expression promotes the progression of hepatocellular carcinoma. It's a topic of interest in the related areas, the bioinformatic-analysis and experiments are proper and comprehensive. But the paper requires some improvement before acceptance.

Line 234, the authors declare that shCDCA2-3 has the best knock-down efficiency according to Figure 1I. However, Figure 1I shows that CDCA2 protein level of Huh7-sh CDCA2-3 is the most similar to Huh7-shNC, which means shCDCA2-3 has the least knock-down efficiency.

There are some mistakes in this manuscript:
1. Line 211 and title in Table 1, “CDCD2” should be corrected as “CDCA2”.
2. Line 230, “L02 cell lines” should be corrected as “L02 cell line”.
3. Line 273-277, the spelling of “Cdk” should be capitalized as “CDK”. Several nonstandard spelling issues distribute in the whole text.
4. The legend of Figure 2(A) is obviously wrong.

The English writing should be improved.

Experimental design

no comment

Validity of the findings

no comment

Reviewer 3 ·

Basic reporting

In the title is using the word "mannar" it is probably a mistyping
check the word it is probably refering to "manner"
I suggest to check grammar in some lines that could be improved. i.e. 274,275 lines: "and found that the expression of CDCA2" it could be better to use "and found that CDCA2 expression". Some parts in the text could be improved.

Experimental design

a) p21 primers sequence is not making correct aligment it is nonspecific, can you check the sequences. It is probably just a typping error.
It is missing one nucleotide in the reverse, Forward is 20 nucleotides and reverse 19, current sequence is not maching with p21 gene
b) Why did you choose to use 2 different test to compare ratios? which criterium did you use to choose?

Validity of the findings

no comment.

Additional comments

Results concerning figure 1.-
In line 234,235
It is stated that shCDCA2-3 have the best efficiency but western blot shows the opposite it is very necessary to clarify this
since it is used for subsequent studies.


Results concerning table 1
Important issue to check:
In line 213 it is stated that high expression is related to older age but table 1 percentajes do not match
it is showed 106 samples = 32.6 % (low) and 83 samples = 67.4% (high)
in this case table 1 shows the opposite people <= 60 shows higher expression


Results concerning figure 2
Is it already know that E-cadherine is absent in SK-Hep1 cells? could you give a source of that information?
or why is it absent in your experiment?


Results concerning figure 4
why did you choose to show in C 20 uM for HepG2 instead of 50 uM? according to letter D 50 is the best concentration

---

## Round 0.2 · accepted · Accept

Thank you for the careful corrections to the manuscript. I am happy to now recommend acceptance.